# Three-Dimensional Bioprinting of Organoid-Based Scaffolds (OBST) for Long-Term Nanoparticle Toxicology Investigation

**DOI:** 10.3390/ijms24076595

**Published:** 2023-04-01

**Authors:** Amparo Guerrero Gerbolés, Maricla Galetti, Stefano Rossi, Francesco Paolo lo Muzio, Silvana Pinelli, Nicola Delmonte, Cristina Caffarra Malvezzi, Claudio Macaluso, Michele Miragoli, Ruben Foresti

**Affiliations:** 1Department of Medicine and Surgery, University of Parma, 43126 Parma, Italy; 2Department of Occupational and Environmental Medicine, Epidemiology and Hygiene, Italian Workers’ Compensation Authority-INAIL, 00078 Rome, Italy; 3Department of Engineering and Architecture, University of Parma, 43124 Parma, Italy; 4Humanitas Research Hospital, IRCCS, 20089 Milan, Italy; 5CERT, Center of Excellence for Toxicological Research, 43126 Parma, Italy; 6CNR-IMEM, Italian National Research Council, Institute of Materials for Electronics and Magnetism, 43124 Parma, Italy

**Keywords:** 3D bioprinter, nanoparticles, long-term culture, nanotoxicology, Calu-3

## Abstract

The toxicity of nanoparticles absorbed through contact or inhalation is one of the major concerns for public health. It is mandatory to continually evaluate the toxicity of nanomaterials. In vitro nanotoxicological studies are conventionally limited by the two dimensions. Although 3D bioprinting has been recently adopted for three-dimensional culture in the context of drug release and tissue regeneration, little is known regarding its use for nanotoxicology investigation. Therefore, aiming to simulate the exposure of lung cells to nanoparticles, we developed organoid-based scaffolds for long-term studies in immortalized cell lines. We printed the viscous cell-laden material via a customized 3D bioprinter and subsequently exposed the scaffold to either 40 nm latex-fluorescent or 11–14 nm silver nanoparticles. The number of cells significantly increased on the 14th day in the 3D environment, from 5 × 10^5^ to 1.27 × 10^6^, showing a 91% lipid peroxidation reduction over time and minimal cell death observed throughout 21 days. Administered fluorescent nanoparticles can diffuse throughout the 3D-printed scaffolds while this was not the case for the unprinted ones. A significant increment in cell viability from 3D vs. 2D cultures exposed to silver nanoparticles has been demonstrated. This shows toxicology responses that recapitulate in vivo experiments, such as inhaled silver nanoparticles. The results open a new perspective in 3D protocols for nanotoxicology investigation supporting 3Rs.

## 1. Introduction

The massive expansion of and demand for engineered nanomaterials (ENMs) [1], from industrial to pharma applications, is facing an increasing trend in this period. As a result, ENMs are becoming a persistent component of the air we breathe [2] and of inhaled therapies, exploiting nanoparticles to intentionally administer drugs [3,4,5]. Hence, inhaled and contact ENMs are becoming a contemporary major public health concern, as workers and the entire population [6,7] are daily exposed to these new materials, through direct skin interaction [8], mucous membranes or by simply breathing [9]. Therefore, nanotoxicology investigations [3] on the effects/problems caused by ENMs over time are mandatory.

Nanomedicine and nanotoxicology are both emerging fields of study, covering basic, translational and clinical science [10,11]. Cell lines are frequently used for in vitro nanotoxicology investigation, but studies are limited to the 2D cell life/morphology and passages [12], with the kinetics of NP internalization [13] and oxidative stress affecting bi-dimensional cultures. A huge range of antioxidants showed promising effects in vitro that could not be recapitulated at the organ level, suggesting that cell culture may adopt reactive-oxygen-species (ROS)-dependent signal transduction that never or minimally operates in vivo [14]. Moreover, several chemotherapies shown to have an effect in 2D culture do not recapitulate the same effect in vivo [15,16,17], and the long-term maintenance (i.e., activity and phenotype [18]) of fully differentiated primary cells [3,19,20] is a challenge.

In vitro nanotoxicology approaches (conventional two-dimensional (2D) cell-based models) are effective for the investigation of the nanoparticles–cell interaction, kinetics and the related cell behavior [21], but, having a single layer of cells completely exposed to the environment, may not recapitulate adequately the actual in vivo interaction. This limitation in the lack of a Z-dimension excludes the possibility of examining NP diffusion, distribution, penetration and the ability to mimic some properties of normal and pathological tissues. Moreover, cell lines are time-/passage-dependent, and several primary cell lines do not survive in culture for more than 12–24 h [22] or cannot even be cultured on conventional cell plates [23].

Insects [24], rodents, rabbits and pigs are widely used for in vivo nanotoxicology and nanomedical preclinical studies [3,25,26,27], with advantages in understanding bioavailability and biodistribution among organs, but are not exempt from limitations, not only related to animal/facility costs, such as the evaluation of the number of nanoparticles interacting with cellular populations and of the kinetics and diffusion of nanoparticles–membrane internalization.

Toxicology and nanomedicine studies require the mandatory step of in vivo investigations [7,28]. This is owed to the need to fully understand the bioavailability, fate and biodistribution of nanoparticles during and after exposure, as well as the local and systemic effects they exert.

The aforementioned step may be largely mitigated by using organoids [29] that reproduce the complex microarchitecture of extracellular matrix (ECM) components and multiple cell type interactions in a sufficient way to recapitulate biological functions [30], consequently reducing the number of animal experiments necessary for toxicology/pharmacological preclinical assessment [31].

The fabrication of organoids can be implemented with new technologies including additive manufacturing [32] that, by using a 3D bioprinter [33], initiates an important innovation in the context of in vitro tissue regeneration [34] and investigation [35], by mimicking the in vivo environment in both mono- and multicellular culture [36,37]. Extrusion-based 3D bioprinting (robocasting or direct ink writing technologies) promises to obtain cell-laden scaffolds based on biocompatible hydrogels accomplishing fast, sterile and reproducible processes [38,39]. However, very little investigation has been performed for nanotoxicology analysis using 3D printer cell-seeded or cell-laden scaffolds, possibly due to the complexity of reproducing and controlling bioprinted “live” multilayers [40,41].

This study, thanks to a customized and cost-effective 3D bioprinter, aims to evaluate the advantages and limitations of bioprinted cell-laden hydrogel scaffolds [42] for nanotoxicology and nanomedical studies or OBST (organoid-based scaffolds for toxicological investigation).

We evaluated alginate/gelatin/Matrigel-based hydrogels at different concentrations, to select the most capable to preserve cell viability and print the cell-laden scaffolds with a honeycomb conventional trajectory [43]. In the second phase, cell viability was analyzed and the results showed that inside OBST, cells were able to grow for 21 days without substantial intervention from the operators, demonstrating that our hydrogel composition protects cells over time, showing a reduction in lipid peroxidation. Finally, we analyzed the added nanoparticles on the OBST, identifying a diffused interaction with the bioprinted cells, generating a toxicology response similar to the in vivo tests breathing AgNPs. Atoxic carboxyl-modified fluorescent nanoparticles (D = 40 nm) were employed for imaging the distribution within the OBST via two-photon microscopy [27] while AgNPs were employed for their well-known cytotoxicity [44,45,46].

The proposed OBST technique shows several advantages for nanotoxicology/nanomedical investigation: (i) cells can survive longer without passages, (ii) nanoparticles can spread and diffuse in the cell-laden multilayer by mimicking the in vivo exposure and (iii) nanoparticles reach the 3D-printed cells in all layers with a noticeable increment of internalization time compared to the unprinted and conventional 2D cultures, iv) different dose/response of 3D-printed cell multilayer toward silver nanoparticles (AgNPs) vs. 2D, resembling in vivo data in zebrafish [45], in insects [47] and rodents [48,49].

## 2. Results

### 2.1. Hydrogel Composition for Preserved Geometries

The developed layer-by-layer structure was made by porcine skin gelatin (10%) enriched with Matrigel^®^ (10%), modulating the percentage of sodium alginate (SA). Then, the final composite hydrogel was selected from different SA concentrations and tested to achieve the best fitting between technological and biological parameters.

In detail, the minimum viscosities were preserved at 35 °C for the SA concentrations ranging from 1 to 4% while the minimal extrusion force at 35 °C was achieved by all the concentrations under 3% (with an applied force between 2 N and 1.48 N, Figure 1A). The extrusion speed, important for preserving the geometry (Figure 1B) and cell comfort due to the force and the route throughout the needle, reduced linearly during the gradual increment of SA concentration at 35 °C (R^2^ = 0.96), while the correlation between viscosity and alginate concentration at 35 °C fits exponentially with an R^2^ = 0.97. Therefore, to reduce variability in the comparison analyses, perfectly fitting those adopted for 2D cell culture (Figure 2), the SA concentration adopted for the toxicology experiments was 2% (viscosity 900cP, Figure 1C), and the extrusion speed was fixed at 13 mm/s.

### 2.2. Three-Dimensional-Printed Calu-3 Cells Proliferate and Survive Longer in Culture

For the investigation of the Calu-3 cell preservation of proliferative activity after 3D bioprinting, it was necessary to evaluate the absence of cell morphology damages and the related viability inside the scaffold after 21 days of culture (Figure 2).

The fluorescence calcein AM assay (Figure 3A) shows that cell proliferation increases significantly at 14 and 21 days (Figure 3B, green line): the number of live cells was 5 × 10^5^ ± 5 × 10^2^ at the beginning of culturing and increased to 1.27 × 10^6^ ± 9.19 × 10^4^ after 21 days. Moreover, the number of dead cells was low at each time point, ranging from 4.78 × 10^4^ ± 7.95 × 10^3^ on Day 0 to 8.25 × 10^4^ ± 2.19 × 10^4^ on Day 21 (*p* = 0.044). 

### 2.3. Oxidative Stress Induction over Time in 3D-Bioprinted Calu-3-Laden Multilayer

Aiming to evaluate the jeopardization of cell-laden multilayer cell activity, the oxidative stress, associated with membrane lipid peroxidation, was assessed by quantification of TBARS levels. Results showed, after 1, 7 and 14 days of 3D-culture printing (Figure 4), a significant and progressive decrement of TBARS values (calculated as μmol/number of cells) obtained immediately after bioprinting (normalized vs. T0). The value of 0.64 × 10^−6^ μmol/ N cells (100%) after 24 h decreased by 91% after 14 days of 3D-culture conditions (Figure 4).

### 2.4. Cell Cycle after 3D Printing

3D printing affected the cell cycle during the first few days after printing in this new culture condition (1–3 days). By flow cytometric analysis, after multilayer solubilization by ethylenediaminetetraacetic acid (EDTA), the Calu-3 cell cycle profile was taken into consideration (Figure 5). In the conventional 2D culture, 63.3% of cells were in the G0/G1 phase, 18.2% in S and 18.1% in the G2/M phase. Just after printing, we observed similar distribution as observed in 2D: 65.2% in G0/G1, 14.7% in S and 20.10% in the G2/M phase. This distribution remained unchanged after 24 h. Interestingly, after 72 h, cells were mainly in the G0/G1 phase (80.5%), while 10.5% were in S and 8.9% in the G2/M phase (Figure 5A,B, Appendix A).

### 2.5. Nanotoxicology Investigation in 3D-Bioprinted Cells

We aimed to understand the relationship between nanoparticle (NP) penetration and OBST biophysical properties. Then, once we confirmed that printed cells embedded in the hydrogel could survive and proliferate over time by drastically reducing oxidative stress, we decided to further investigate the potential use of our “organoids” for nanotoxicology approaches, analyzing the NP (red fluorescent 40 nm latex, negatively charged [27,50]) internalization in 2D and 3D multicellular structures, printed or not. The two-dimensional Calu-3 cell line internalized NPs in ca. 30 min, while the 3D-bioprinted Calu-3 cells did so in ca. 24 h (Appendix A).

After 48 h, the fluorescent NPs deeply penetrated the unprinted (2 mL “just-dropped” hydrogel) and printed (2 mL hydrogel) multilayer with different results (Figure 6). In the unprinted one, although the NPs reached the cells (Figure 6A), the bulk denoted air bubbles within the printed object. Moreover, the cells were narrowly dispersed between 200 and 480 μm thickness and the administered “on top” NPs penetrated the multilayer up to 500 µm (Figure 6D). On the contrary, we did show a direct interaction between nanoparticles and the cells within the OBST (Figure 6B,C). The cells were widely distributed in the entire thickness of the multilayer and for this type of cell-laden bioprinted hydrogel, the maximal penetrated nanoparticle numbers were achieved at Z = 680 µm (Figure 6E).

Finally, we evaluated the polydispersity of colloidal toxic AgNPs, known as a potential agent for possible antitumor therapies [49]. Two-dimensional-cultured Calu-3 cells were treated with increasing concentrations of AgNPs and we calculated the percentage of cell death after three days of treatment. The AgNP concentration used (100 µg/mL) denoted a 30% inhibition of cell viability (Figure 7A) with 50% inhibition (IC_50_) calculated at 2.07 × 10^2^ ppm, similar to what was detected in the in vitro toxicological investigation on zebrafish [45]. On the contrary, for all AgNP concentrations tested in 3D-culture conditions, the percentage of viable cells denoted an increment of the concentration necessary to inhibit 50% of cell viability (IC_50_ = 6.88 × 10^9^) (Figure 7B), similar to in vivo toxicological investigation on rats, where silver concentration became toxic with concentrations over 100 µg/mL [46,48].

## 3. Discussion

Our technology aims to produce OBST devices capable of reducing, refining and replacing a significant proportion of animal tests for toxicology. To assess nanotoxicology investigations via 3D-printed cell-laden scaffolds that mimic real in vivo cell–cell intermingling [51] and the production of extracellular matrix [52,53], which can hardly be observed in 2D, we designed and assembled a 3D bioprinter. The 3D bioprinter was designed and assembled to minimize the possibility of contamination during hydrogel extrusion using minimal extrusion force and speed with a well-known concentration of SA [54].

Cell lines expanded in 2D were included with their specific supplementary media into the alginate/gelatin-based hydrogel formulation [55], which was also supplemented with Matrigel^®^ [56], and the final computer-aided design (CAD) drawings based on bio-inspired honeycomb crisscrossing layers [43] defined the trajectory for the OBST fabrication.

All the printed cells were biologically active, overcoming 21 days with minimal operator interventions, and able to internalize nanoparticles, subsequently added on the OBST to simulate skin and mucous membrane interaction with ENMs. Moreover, the thiobarbituric acid reactive substances (TBARS) level significantly decreased by ca. 90% during 14 days in culture for Calu-3 (cf. Figure 4), suggesting that the hydrogel composition may protect the 3D-printed cells for long-term investigation. TBARS indicated that a 3D environment re-adaptation time was necessary for all the cell lines tested and, as previously reported for cell proliferation, after a short period, cells re-activated their normal biological functions, returning to a reduced membrane lipid peroxidation condition. This re-adaptation was also confirmed by the cell cycle analysis; our data showed a reduction in the number of cells in the G2/M phases 72 h after printing compared to time 0, and an increase in the number of cells in the G0/G1 phase. In our experimental conditions, cells printed in the 3D OBST began to replicate after 7 days (see Figure 3), reaching a significant increase in cell number after 14 days. The number of dead cells remained minimal over this period. It was also important to ensure uniform cell deposition when producing an OBST; unprinted scaffolds displayed not only non-uniformly distributed cells, but also air bubbles and non-uniform internalization of NPs by the OBST.

Finally, it is interesting to note that while the 2D in vitro toxicity of high doses of AgNPs is undisputable for mammal cells [57,58], the OBST reached results able to resemble in vivo toxicology effects on rats after AgNP (11–14 nm) inhalation. Moreover, in our study, we obtained a less marked reduction in 3D-printed cell viability exposed to AgNPs and the possibility to investigate the same cells embedded in a 3D architecture for weeks may certainly accelerate the bench-to-bedside process. Cell-laden 3D-printed multiple layers allow the possibility to investigate lipid peroxidation longitudinally, as the damage produced by oxidative stress is comparable to lung tissue studies in vivo [28] and, depending on nanoparticles’ physicochemical characteristics [50], they can be pursued for toxicological or nanomedical study, as nanoparticles can diffuse successfully within the printed layers.

The indication that 3D OBSTs do not require cell passages and can survive in culture for >21 days may suggest adopting 3D printing for long-term investigation in both field nanotoxicology and nanomedicine. While the OBST technique is still in its infancy and requires further investigations, it may become an effective tool for nanotoxicology investigations where the cells embedded in the 3D hydrogel are active and interact with the NPs [59] and the scaffold [60].

## 4. Materials and Methods

### 4.1. 3D Customized Bioprinter

The 3D bioprinter was customized in our laboratory as recently published [39]. Briefly, the Cartesian aluminum skeleton 47 × 37 × 47 cm was designed for the best fitting between the 3D printer and the dedicated biosafety cabinet (Heraeus, Herasafe, Hanau, Germany), provided with UV light for the sterilization procedures before printing. Transmission belts assure an x,y micrometer movement of the temperature-controlled printing platform, where the Petri dish can be placed and secured during the printing process, by using stepper motors (hybrid rotor, NEMA size 17, 2.8 kg/cm). Moreover, the same motors were adopted to equip the printer with double temperature-controlled micromechanical extruders (two cylindrical aluminum coils surrounded by resistors), to manage the hydrogel deposition via two disposable syringes (5 mL, BD Emerald, Milano, Italy; nozzle: 21G sterile needle).

### 4.2. 3D Model Design

The multilayer 3D model was made via SolidWorks 2015 and saved as a standard triangulation language (.stl) file, for the trajectory generation with Slic3r™ (https://slic3r.org/). Then, the aforementioned file was converted into g-code format, useful for the customized firmware (Marlin™) and, consequently, motion control. Hence, the geometry regularity of the OBST (variability between digital and printed model) was measured using an inverted epifluorescent microscope (Nikon, Eclipse 2000, Balsamo Strumenti, Medicina, Italy) and the pictures of the scaffold were acquired using a microscope camera (CCD Sony Corporation, Shinagawa, Tokyo, Japan). Finally, the expanded cells, included in the bioink, were seeded following the CAD drawing (from 7 to 21 crisscrossing sinusoidal 180° phase-shift layers (honeycomb)) resulting in a defined 3D assembly of 16 × 16 × 1.5 mm.

### 4.3. Cell Line Culture

Human non-small cell lung cancer line Calu-3 (Calu-3), purchased from American Type Culture Collection (ATCC, Manassas, VA, USA), was cultured in D-MEM (high glucose, Thermofisher, Monza, Italy) with 0.01 mmol/L nonessential amino acid (Thermofisher, Monza, Italy), 1 mM sodium pyruvate (Merck, USA), supplemented with 2 mM glutamine (Thermofisher, Monza, Italy), 10% fetal bovine serum (FBS, Euroclone, Milano, Italy) and 1% penicillin/streptomycin (5000 U/mL, Thermofisher, Monza, Italy). Then, the cultured cells were maintained under standard and recommended cell culture conditions, at 37 °C in a water-saturated atmosphere of 5% CO_2_ in air, and culture media were changed every three days onto 2D cell culture. Instead, in 3D cell culture, the medium is an integral part of the multilayer, and was not replaced, but, when needed, 0.5–1 mL of supplemented media were added to avoid dehydration.

### 4.4. 3D-Printed Cell-Laden Multilayer

The bioinks were composed by mixing SA (Sigma Aldrich, St. Louis, MO, USA) at different concentrations (1% to 5%), 10% Matrigel^®^ (Tewksbury, MA, USA) and 10% porcine gelatin (Type A, Sigma Aldrich, St. Louis, MO, USA), diluted in 2 mL of Dulbecco’s modified Eagle’s medium (DMEM, Thermofisher, Monza, Italy) (w/o phenol red), the complete medium supplemented with 10% fetal bovine serum (FBS), supplemented with 10 mM HEPES buffer, 2 mM glutamine and 1% penicillin/streptomycin (all from Thermofisher, Monza, Italy). Then, the hydrogel viscosities were measured using a viscometer (PCE-RVI 2, PCE Iberica S.L., Calle Mayor, Tobarra, Spain) and, subsequently, 10 × 10^6^ cells were added at a total volume of 2 mL charged into the disposable syringe (5 mL, BD Biosciences, Franklin Lakes, NJ, USA). Finally, the OBTS was fabricated by printing the cell-laden hydrogel (2–2.5 M cells/layer) under the biosafety cabinet and the extruder deposition force was measured with an extensometer (PCE-FB1k, PCE, Meschede, Germany). Afterward, the hydrogel was crosslinked for 5′ with 50 mM CaCl_2_ solution, washed three times with PBS to remove residual CaCl_2_ solution and, before incubation, a proper amount of complete medium was added.

### 4.5. Cell Proliferation and Viability Analysis

Each hydrogel-containing cell was dissolved with 0.25% Trypsin-EDTA (Thermofisher, Monza, Italy) solution for 20 min at 37 °C in 5% of CO_2_, then trypsin was inactivated adding an equal volume of completed culture medium and the recovering cells from hydrogel were centrifuged at 1200 rpm for 4 min at room temperature. Cell proliferation was quantified by cell counting in a Bürker hemocytometer by trypan blue exclusion (technical triplicate for each multilayer) and evaluated under a phase-contrast upright microscope (Leica Microsystems, Wetzlar, Germany).

The viability of cells in the 3D construct was also confirmed using calcein AM staining (Thermo Fisher, Waltham, MA, USA). Specifically, the 3D construct was maintained in a cell incubator for 1, 7, 14 and 21 days and, at each time point, fresh media containing 5 µM calcein AM was added. After 1 h of incubation at 37 °C in 5% of CO_2_, the construct was washed twice with PBS before being imaged with an upright microscope (Leica Microsystem, Leica Microsystems, Wetzlar, Germany) through a ×20/0.7 or ×40/1.3 oil objective.

The cell cycle was determined by propidium iodide (PI, Thermofisher, Monza, Italy) staining, after dissolving each hydrogel with EDTA solution for 20 min at 37 °C in 5% of CO_2_.

After overnight incubation at 4 °C, samples were washed and incubated with 1 mL of PBS containing 20 μg/mL PI and 12.5 μL µg/mL RNase (1 mg/mL in water), then stained cells were analyzed and sorted by an FC500™ flow cytometer (Instrumentation Laboratory, Bedford, MA, USA). At least 20,000 events were counted. The percentages of cells occupying the different phases of the cell cycle were calculated by FlowJo Software (Tree Star Inc., Ashland, OR, USA).

### 4.6. Oxidative Stress Assay

After dissolving each hydrogel in 0.25%Trypsin-EDTA solution for 20 min at 37 °C in 5% of CO_2_, intracellular ROS levels were quantified by flow cytometer using 2′,7′-dichlorodihydrofluorescein diacetate (DCFH-DA). Moreover, the well-established TBARS method was detected to monitor lipid peroxidation, as already described [28], using a Cary Eclipse fluorescence spectrophotometer (Varian, Inc., Palo Alto, CA, USA) (excitation 515 nm, emission 545 nm). All values were normalized for the cell number.

### 4.7. Nanoparticles Diffusion and Multilayer Analyses

Cells were incubated with a lipophilic membrane stain (1,1’-Dioctadecyl-3,3,3’,3’-Tetramethylindocarbocyanine Perchlorate, DiI, Thermofisher, Monza, Italy) with an emission of 580 nm, before being printed. We administered 20 µg/mL yellow/green-fluorescent nanoparticles on top of the three-day-old multilayers and incubated them for 48 h before fixation (PFA 4%). Doses were selected from previous experience with these NPs [27,50]. Images of the assembly containing Calu-3 cells with 20 µg/mL nanoparticles administration were performed utilizing two-photon microscopy (res. 0.5–1 µm) at 25× magnifications (Nikon, BalsamoStrumenti, Medicina, Italy). Three-dimensional imaging, nanoparticle diffusion quantification and rendering were performed with IMARIS 7.6 (Bitplane AG, Schlieren, Switzerland). Green/yellow and far-red nanoparticles with a maximal excitation spectrum of 515 and 580 nm were used in this work. Forty-nanometer, latex carboxylate FluoSpheres™ were purchased from Molecular Probes (Thermofisher, Monza, Italy, Code F10720). Customized silver colloidal nanoparticles (AgNPs, 1000 ppm stock solution) with a diameter of 10–20 nm were kindly provided by Q Bio System Limited (UK). Dose/response was performed vs. cell viability from 0 to 1000 ppm. All nanoparticles were sonicated for 30 min at T = 37 °C before being used.

### 4.8. Statistical Analysis

Statistical analyses were carried out using GraphPad Prism version 6.0 software (GraphPad Software Inc., San Diego, CA, USA). Data were normally distributed and expressed as mean ± SD (standard deviation). Normal distribution was checked by the Kolmogorov–Smirnov test. Differences between the mean values recorded for different experimental conditions were evaluated by Student’s *t*-test or by one-way ANOVA followed by Bonferroni’s post-test, and *p* values are indicated where appropriate in the figures and their legends. A value of *p* < 0.05 was considered significant.

## 5. Conclusions

Our study demonstrates the potential of 3D bioprinting as a promising technology for nanotoxicological investigation. The unique benefits of this novel additive manufacturing technology for cell culture include the long-term culturing of cell lines due to the reduction in oxidative stress over time, the viability of 3D-printed cells once embedded in the hydrogel and the absence of time-passages from the operator. These benefits allow for a more accurate and comprehensive investigation of the effects of nanoparticles on living cells, as well as the potential for more efficient and cost-effective experimentation. Additionally, our research highlights the significant dissimilarities between 2D vs. 3D data, which suggest the need for a revisiting tactic in the fields of nanotoxicology and nanomedicine to account for potential effects on cell morphology and cell–cell interaction in a 3D environment. Ultimately, we believe that our technology can contribute to the development of safer and more effective nanomedicine, as well as provide a valuable tool for researchers in the field of nanotoxicology.

## 6. Limitations

This technology, although very promising, is not exempted from limitations. The number of seeded cells strictly depends on the 3D printing parameters and is normally less than the number of cells embedded in the hydrogel-loaded syringe; this is due to the residual volume in the syringe Luer lock and the force exerted during the extrusion that, although kept minimal, mechanically disrupts the cells.

## Figures and Tables

**Figure 1 ijms-24-06595-f001:**
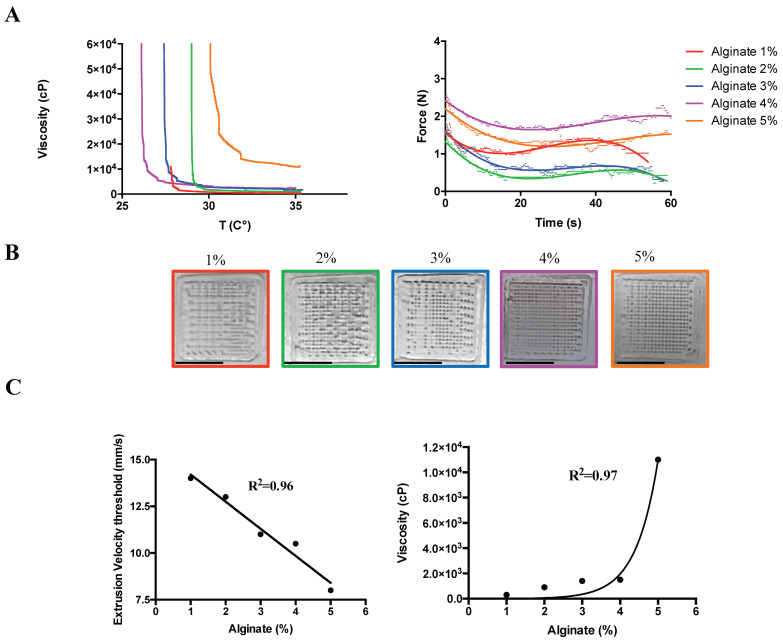
Three-dimensional bioprinting process parameters. (**A**) Left: hydrogel viscosity calculated at different sodium alginate (SA) concentrations for different extrusion temperatures (25–37 °C). Right: extrusion force measured for the different SA at 35 °C. (**B**) Preserved geometry at different SA: from left to right: 1, 2, 3, 4 and 5%. Scale Bar: 10 mm. (**C**) Left: minimal extrusion speed threshold able to preserve the geometry in (**B**) at different SA concentrations. Right: hydrogel viscosity calculated at the minimal extrusion speed for the different SA concentrations at 35 °C.

**Figure 2 ijms-24-06595-f002:**
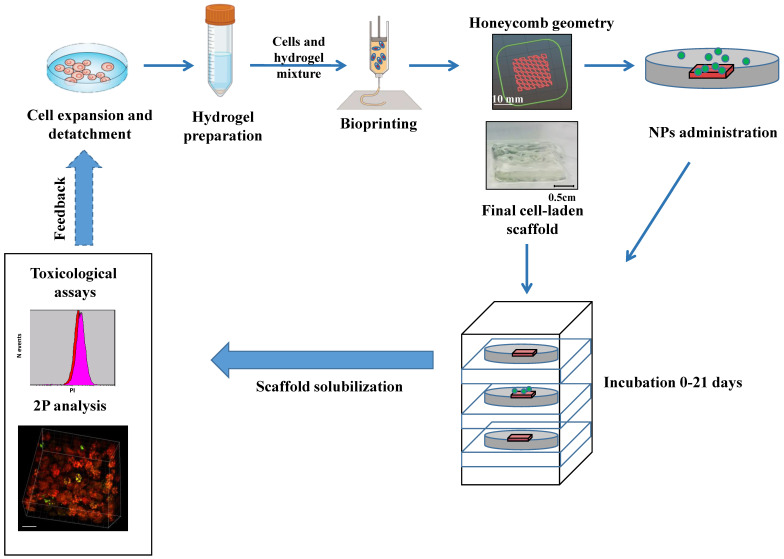
Study overview and experimental protocol. Experimental protocol for cell-laden multilayer generation, imaging and toxicological assay. Feedback was adopted to better define the 3D printing process parameters. NPs: carboxyl fluorescent nanosphere or polydispersity colloidal silver nanoparticles, 2P: two-photon microscopy. Modified from www.BioRender.com.

**Figure 3 ijms-24-06595-f003:**
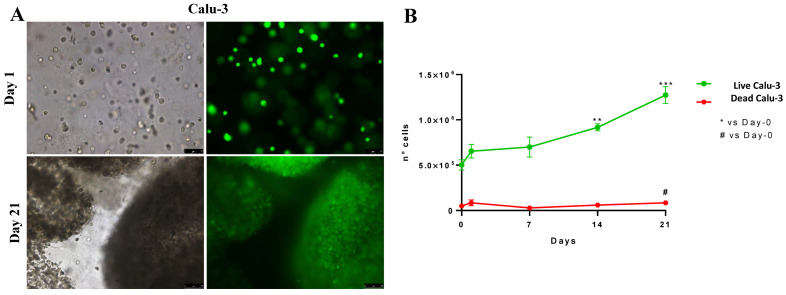
Cellular viability in the 3D-printed cell-laden multilayer. (**A**) Brightfield (left) and calcein-AM-loaded (right) cells at day 1 and day 21, respectively. (**B**) Cell viability measured at day 1, 7, 14 and 21 in 3D culture. N = 8 for each time-point. Significance set as ** *p* < 0.01, *** *p* < 0.001 for live cells (green line) and # *p* < 0.05 for death cells (red line) vs. control at time 0 (Day-0). Data are expressed as mean ± SD. Scale bars: 1 mm.

**Figure 4 ijms-24-06595-f004:**
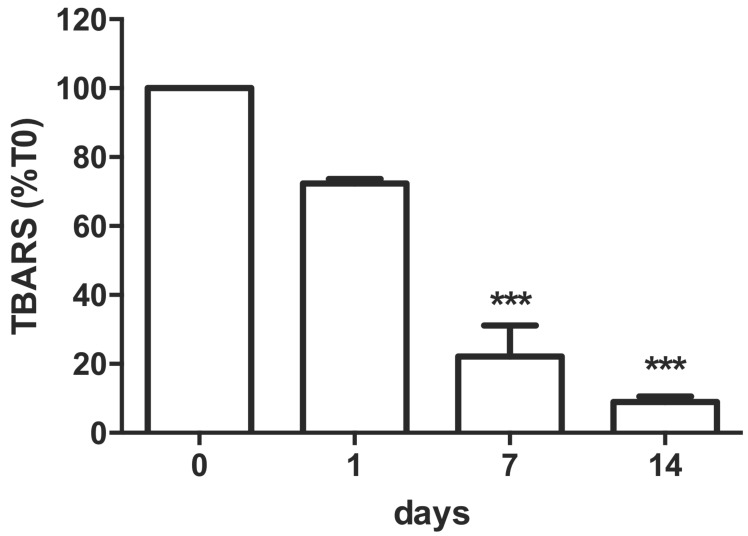
Thiobarbituric acid reactive substance assays (TBARS). TBARS for printed Calu-3 cells after hydrogel solubilization after 1 h (T0), and 1, 7 and 14 days in culture. *** *p* < 0.001 vs. T0. Data are expressed in %mean ± SD vs. T0, represented as percentage vs. control. N = 4.

**Figure 5 ijms-24-06595-f005:**
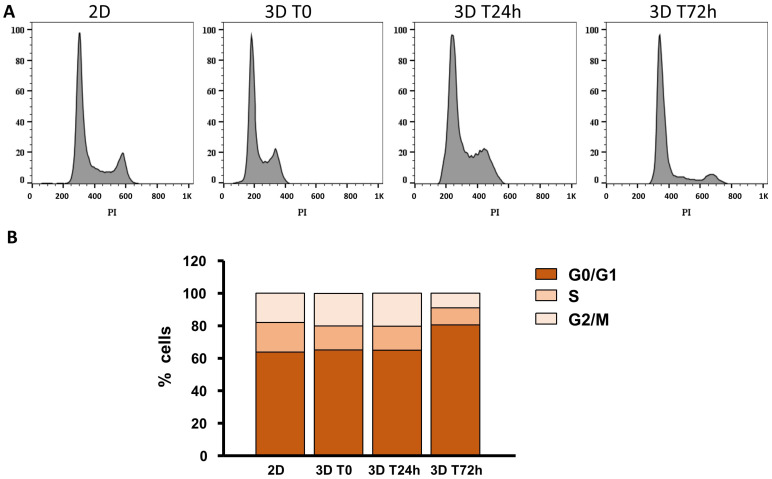
Cell cycle in unprinted and printed cells. (**A**) Representative cell cycle obtained by FACS for Calu-3 cells in 2D culture and 3D-printed at Time 0, after 24 and 72 h in culture, respectively. (**B**) Cell cycle for the culture obtained in (**A**) phases: G0/G1, S, G2/M. PI: propidium iodide (n = 3).

**Figure 6 ijms-24-06595-f006:**
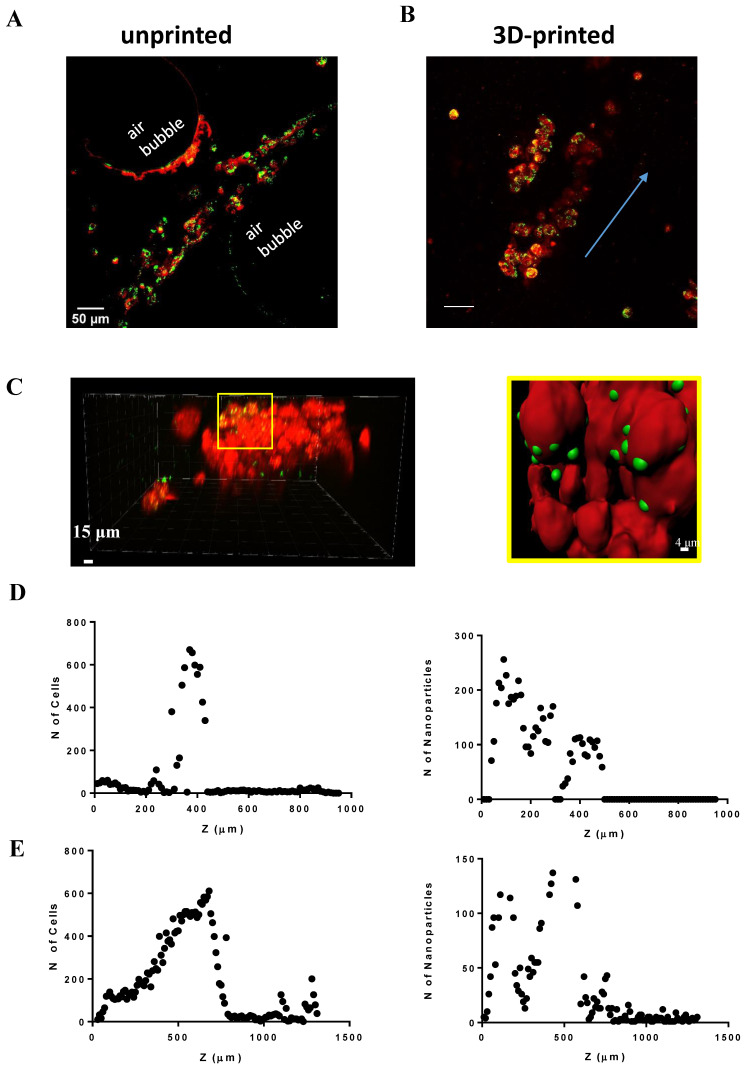
Nanoparticle diffusion in the uncontrolled deposition (unprinted) and printed OBST. Microscopical images on unprinted (**A**) and printed (**B**) hydrogel showing NP (green) and Calu-3 (red) interaction, respectively, in cell-laden multilayers (from top (0) to bottom (1500 µm)). Blue arrow: cell growth direction. (**C**) Left: a portion of a 3D-printed multilayer. Right: rendered image of the yellow zoom shown in the left panel. NP internalization from the cells. (**D**) Left: cell distribution in the unprinted hydrogel. Right: NP diffusion in the unprinted hydrogel. (**E**) Same as (**D**) for 3D-printed hydrogel.

**Figure 7 ijms-24-06595-f007:**
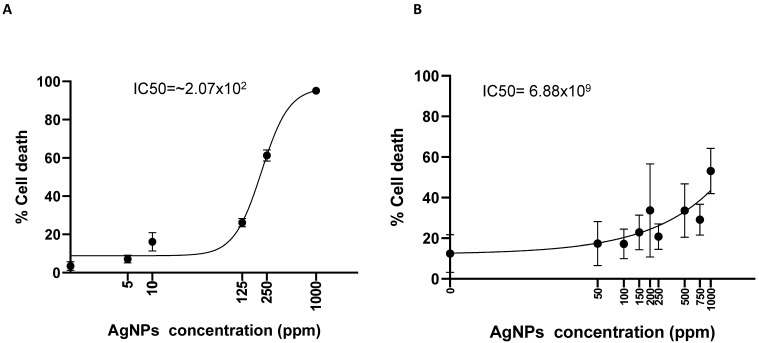
Nanotoxicity exerted by silver nanoparticles in 3D-printed cell-laden multilayer. (**A**) Dose/response for Calu-3 exposed to Ag NPs (from 0 to 1000 ppm) in 2D environment for 48 h. (**B**) Same as (**A**) for the 3D-printed cells.

## Data Availability

Data analysis will be available upon reasonable request.

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
