# Peer review of "Three-Dimensional Bioprinting of Organoid-Based Scaffolds (OBST) for Long-Term Nanoparticle Toxicology Investigation"

_ijms, 2023, doi:10.3390/ijms24076595_

Round 1

Reviewer 1 Report

The study highlights the potentiality of 3D bioprinting in generating organoid-like scaffolds that can be used to investigate the toxicity response when exposed to metallic nanoparticles (AgNPs). Overall, the Idea is interesting. Therefore, I recommend the publication of this work with minor corrections.

1.      While the introduction is well-written, the authors stated in the abstract that they employed two classes of nanomaterials, latex-fluorescent or smaller AgNPs; however, not clearly mentioned, in the introduction, the motivation behind their selection.

2.      The authors ought to follow the order of the journal by having the section of materials methods before results and discussion. Figure 2 is not cited in the text. Thus, I suggest the authors to cite it along when elaborating the experimental work in material and method section.

3.      The heading 2.1 is long, authors are encouraged to re-name the sections with less words.

4.      The quality of figure 1 is extremely low. The letters are not visible.

5.      2.2: please correct “2.2. 3D”

6.      Please add the full description for all abbreviations in their first mention in the text.

7.      Revise the whole text over again as some parts are not in well-coordination. (e.g., sections 2.4 and figure 4.)

8.      The error bars are missing in Figure 4 for day 1.

9.      Line 153 : re-write the phrase starting from (substantial no change..)

10.  Please rewrite the caption in figure 6.

11.  There is a lot of redundancy in the discussion part. This should be merged with the results

Author Response

The study highlights the potentiality of 3D bioprinting in generating organoid-like scaffolds that can be used to investigate the toxicity response when exposed to metallic nanoparticles (AgNPs). Overall, the Idea is interesting. Therefore, I recommend the publication of this work with minor corrections.

We thank the reviewer for her/his valuable suggestions aimed at improving our manuscript

Q 1. While the introduction is well-written, the authors stated in the abstract that they employed two classes of nanomaterials, latex fluorescent or smaller AgNPs; however, not clearly mentioned, in the introduction, the motivation behind their selection.

A1 We agree with the suggestions, and we modified the abstract and the introduction accordingly.

Page 1: “Administered fluorescent nanoparticles can diffuse throughout the 3D printed scaffolds while this was not the case for the unprinted ones.

A significant increment in cell viability from 3D vs 2D cultures exposed to silver nanoparticles has been demonstrated. This show toxicology responses that recapitulate in-vivo experiments, like inhaled silver nanoparticles.

Line 105, Page 3: Atoxic carboxyl-modified fluorescent nanoparticles (D=40 nm) have been employed for imaging the distribution within the OBST via two-photon microscopy [26] while AgNPs nanoparticles have been employed for their well-known cytotoxicity [43-45].

Q 2. The authors ought to follow the order of the journal by having the section of materials methods before results and discussion. Figure 2 is not cited in the text. Thus, I suggest the authors to cite it along when elaborating the experimental work in the material and method section.

A 3.  We adopted the IJMS template which includes the methods section after the Discussion section. This is in agreement with last articles published in IJMS. We will contact the Editorial Office for the correct proof. We include the mention for Figure 2 (Line 141, Page 4 and Line 149 Page 5).

Q 3. The heading 2.1 is long, authors are encouraged to rename the sections with less words.

A3 We follow the reviewer suggestion and change the heading in “Hydrogel composition for preserved geometries”.

Q 4. The quality of figure 1 is extremely low. The letters are not visible.

A 4. We apologize for this inconvenience, probably due to the final conversion of the file. We increase the resolution of Figure 1 accordingly.

Q 5. 2.2: please correct “2.2. 3D”

A 5 Done as suggested

Q 6. Please add the full description for all abbreviations in their first mention in the text.

A 6. Done as suggested

Q 7. Revise the whole text over again as some parts are not in well-coordination. (e.g., sections 2.4 and figure 4.)

A 7 Some parts are now re-coordinated. However, because Section 2.2 mentioned two figures the subsequent figures are shifted (Figure 4 in section 2.3, figure 5 in section 2.4 etc.)

Q 8. The error bars are missing in Figure 4 for day 1.

A 8. The reviewer is correct. SD bar has been added

Q 9. Line 153: re-write the phrase starting from (substantial no change..)

A 9 We change the sentence in “This distribution remained unchanged after 24 hrs” (Line 186, Page 6)

 Q 10. Please rewrite the caption in figure 6.

A 10 Done as Suggested

Q 11. There is a lot of redundancy in the discussion part. This should be merged with the results

A 11. We understand the reviewer’s suggestion and we limited the discussion section strictly to the explanation of the results.

Reviewer 2 Report

Comments in the attachment.

Author Response

in this research, the Authors have assessed usefulness of 3D bioprinting of Organoid-Based Scaffolds

for nanoparticle toxicity testing. According to the Authors OBS is a promising support for

nanotoxicological research.

We thank the reviewer for her/his valuable suggestions aimed at improving our manuscript

The study has been described in detail, but there are a few points to consider.

Q1 Figure 1 is completely illegible

A1 We apologize for that. Figure 1 was in a good resolution before the conversion to PDF. We replace Figure 1 with the High-Res version.

Q 2) Line 123 – a typing error “2.23 D”

A2. We rewrote the caption as “2.2 Three-Dimensional printed Calu-3 cells proliferate and survive longer in culture. “

Q3) Line 145/146 – a typing error

A3 Correct.

Q 4) Figure 3 – the legend is unclear (* vs Day 0; # vs Day 0)

A4. The legend refers to the data that are significant vs. Day 0 (Control). We rephrase the legend  as follows: “Figure 3. Cellular viability in the 3D printed cell-laden multilayer. A. Brightfield (left) and Calcein-AM loaded (right) cells at day 1 and day 21 respectively. B. Cell viability was measured at day 1, 7, 14 and 21 in 3D culture. N= 8 for each time-point. Significance set as *p<0.05, **p<0.01, ***p<0.001 for live cells (green line) and #p<0.05, ##p<0.01, ###p<0.001 for death cells (red line) vs. control at time 0 (Day-0). Data are expressed as mean ± SD. Scale bars: 1 mm.”

Q5) Figure 4 – how many independent experiments (TBARS) were performed (N=?) 

A5. TBARS are made by 4 independent experiments and expressed as normalized mean ± SD

Q6) Line 154/155 – a typing error (twice G2/M)

A6 Done as Suggested

Q7 Figure 5 – representative cell cycle is presented but an additional Table with all data

(mean+/-SD; for all n=4) should be included in the manuscript  

A7: We follow the reviewer’s suggestion, and we add the table in the supplementary information file (See new Supplementary Table 1)

Q8 Lack of nanoparticle characteristic. I could not find information about the NP characteristic

even in the references given (line 186 [15,32]). The full characteristic of NP used is essential

when research with NPs is conducted. Are the Author sure that they worked with NPs? Did

they checked agglomeration/aggregation of NPs after suspension (and/or after sonication),

e.g. using DLS in time 0h and after incubation?

A8. We thank the reviewer for highlighting this question. The NPS used in this manuscript are both commercially available. The fluorescent carboxyl-modified 40 nm nanoparticles were previously adopted and described by us in a recent published paper (same stock; see Foresti et al Sci. Rep 2020 https://www.nature.com/articles/s41598-020-60196-y ). Silver Nanoparticles were also commercially available; we tested the dimension with TEM and by Nanosight.

We observed that our AgNPs are in the range of what have been stated within the company datasheet (11-14 nm for airbone diameter), see TEM figures above. We did not use DLS but the Nanosight that reveals hydrodynamic diameters (in our culture media) in the range of 89.5 ± 1.1 nm (Concentration of 3.08x108 particles/ml).

These data just confirm the company data sheet for AgNPs therefore we decided to do not to include it in the manuscript.

Q9) Line 187: Why 2D and 3D cultures internalized NPs at different time (30 min vs 24h)?

A9) This is an important question, and we thank the reviewer for bringing it up. In 2D unprinted culture, all cells are directly exposed to NPs through membrane interaction. We observed a fast internalization of fluorescent NPs, and we published that internalization and toxicity are strictly dependent on the NPs' Z-Potential interacting with the cell membrane (see Miragoli et al., "Functional interaction between charged nanoparticles and cardiac tissue: a new paradigm for cardiac arrhythmia?" Nanomedicine (Lond). 2013 May;8(5):725-737. doi: 10.2217/nnm.12.125). On the other hand, the 3D printed OBST includes cells embedded in the hydrogel, and we observed that NPs' internalization is delayed, probably due to the 3D structure (different size, viscosity). As a result, the penetration time for nanoparticles is also delayed.

Q10) Line 192: “In the unprinted……within the printed object” – what did the Authors mean?

A10) The term "unprinted object" refers to a bioink (consisting of hydrogel and cells) that has been deposited onto a petri dish using a piston-driven air displacement pipette, but without the use of 3D printing technology. In this condition, the cells, hydrogel, and any administered nanoparticles may not be uniformly distributed.

Q11 Line 313-323 – duplicate text (288-298)

A11. We remove the duplicated text. We apologize about that.

Q12 Line 338 – did the Author consider using RNase to measure the cell cycle?

A12. Nuclear DNA content was labelled with propidium iodide (PI). Briefly, about 1×106 cells were harvested, resuspended in PBS (Ca2+ and Mg2+ free and supplemented with EDTA 0.5 mM) and fixed adding 96% cold ethanol. After overnight incubation at 4°C, samples were washed and incubated with 1 ml of PBS containing 20 μg/ml PI and 12.5 μL µg/ml RNase (1 mg/ml in water).

We apologized that we did not include that information in the method section.

We rephrase and add that information in the Methodological section as follows: Page 12 Line 366: “ After overnight incubation at 4°C, samples were washed and incubated with 1 ml of PBS containing 20 μg/ml PI and 12.5 μL µg/ml RNase (1 mg/ml in water), then stained cells were analyzed sorted by a FC500™ flow cytometer (Instrumentation Laboratory, Bedford, MA, USA). At least 20,000 events were counted. The percentages of cells occupying the different phases of the cell cycle were calculated by FlowJo Software (Tree Star Inc, Ashland, OR, U.S.A.). “

Q13) Line 350 – a typing error ?? “as M”

A13 Corrected.

Q14) Line 348 – there is no information about TBARS method in reference 39 – were the results of

TBARS normalized via no. of cells?

A14. We replaced it with the proper reference. Thank you.

Q15 How was the NP concentration used - Lack of Figure 7

A15. We inserted the concentration range in the Legend: “A. Dose/Response for Calu3 exposed to Ag NPs (from 0 to 1000 ppm) in 2D environment for 48 hrs. “

Q16 Inhibition of cell cycle in G0/G1 after 72hwas not discussed by the Author

A16. We agree with the reviewer and we discussed the results in the discussion section (see Answer A17).

Q17 Generally, discussion very poor

A17. The discussion is now more strict to the data. In detail we add new sentences at

Page 10, Line 269: “ This re-adaptation was also confirmed by the cell cycle analysis; our data display a reduction in G2/M phases following the 72 hrs after printing compared to time 0 and an increment in G0/G1 phase. In our experimental conditions, 3D printed cells in the OBST start to replicate after 7 days (cf. Figure 3) reaching a significant increment of cells after 14 days. The number of dead cells remains minimal over this period. It is also important to control cell deposition in producing an OBST; unprinted scaffolds displayed not only undistributed cells but also air bubbles and disomogenities in NPs internalization by the OBST.

We also move part of the discussion in the introduction as not related to the proposed data

Page 02, Line 50. Nanomedicine and nanotoxicology are both emerging fields of study, covering basic, translational, and clinical science [10, 11]. Cell lines are frequently used for in-vitro nanotoxicology investigation, but studies are limited to the 2D cell life/morphology and passages [12], with the kinetic of NPs internalization [13] and oxidative stress affecting bi-dimensional cultures. A huge range of antioxidants showed promising effects in-vitro that could not be recapitulated at the organ level, suggesting that cell culture may adopt reactive oxygen species (ROS)-dependent signal transduction that never or minimal operates in-vivo [14]. Moreover, several chemotherapies shown to have an effect in 2D culture do not recapitulate the same effect in-vivo [15-17], and the long-term maintenance (i.e. activity and phenotype [18]) of fully differentiated primary cells [3, 19, 20] is a challenge.

Reviewer 3 Report

This manuscript investigated the role of 3D bioprinting of Organoid-Based Scaffolds for Long-term Nanoparticles Toxicology Investigation.

I suggest a minor correction and require a detailed clarification. Correction to be addressed by the authors as follows: The abstract is not well organized, where the sentences are incomplete and no continuity is there. It would be feasible, if include the significance of the current study in the abstract. A brief description of how the authors selected information from the literature in the databases, as well as doses.
Authors should justify and expand the information on the biomedical applications , highlighting the main contribution in in vitro fields. Authors should specify the main experimental conditions used on the evidences from the literature. Where they briefly describe the most important data reported in the literature in a homogeneous manner and sequence reinforcing the relevance of green nanomaterials  as medicinal alternative.
The most significant  mechanism of action of Organoid-Based Scaffolds should be described and noticed more emphatically. Authors should discuss whether the use of Organoid-Based Scaffolds represents a solid alternative to existing test systems.
Please add below studies to your manuscript in discussion section and also please discuss about possible role of this system in toxicity investigation of nanomaterials on cell and organells.

DOI: 10.7150/ntno.78611

DOI: 10.1016/j.jksus.2022.102340

DOI: 10.1155/2021/1520052

Conclusions should reaffirm the fundamental contribution of this paper.

Author Response

This manuscript investigated the role of 3D bioprinting of Organoid-Based Scaffolds for Long-term Nanoparticles Toxicology Investigation.

 We thank the reviewer for her/his valuable suggestions aimed at improving our manuscript

Q1 I suggest a minor correction and require a detailed clarification. Correction to be addressed by the authors as follows: The abstract is not well organized, where the sentences are incomplete and no continuity is there. It would be feasible, if include the significance of the current study in the abstract. 

A 1 We agree with the suggestions, and we modified the abstract accordingly.

Q2 A brief description of how the authors selected information from the literature in the databases, as well as doses.

A2. We agree with the reviewer and we add the following information in the text:

Page 3, Line 105 : Atoxic carboxyl-modified fluorescent nanoparticles (D=40 nm) have been employed for imaging the distribution within the OBST via two-photon microscopy [26] while AgNPs nanoparticles have been employed for their well-known cytotoxicity [43-45].

Page 12, Line 384: “We administered 20 µg/ml yellow/green-fluorescent nanoparticles on top of the three-day-old multilayers and incubated them for 48 hrs before fixation (PFA 4%). Doses were selected from previous experience with this NPs[26, 46].

Page 12, Line 393: “Customized silver colloidal nanoparticles (AgNPs, 1000 ppm stock solution) with a diameter of 10-20 nm have been kindly provided by Q Bio System Limited (UK).  Dose/ Response has been performed vs cell Viability from 0 to 1000 ppm.”

Q3. Authors should justify and expand the information on the biomedical applications, highlighting the main contribution in in vitro fields. Authors should specify the main experimental conditions used on the evidences from the literature.

A3. Done as suggested. In details we expand the introduction (Page 2 , Line 50) and the discussion section (Page 9, Line 250 and Page 10, Line 269)

Q4. Where they briefly describe the most important data reported in the literature in a homogeneous manner and sequence reinforcing the relevance of green nanomaterials as medicinal alternative.

A4. Was not the intention of this manuscript to reinforce the relevance of green nanomaterials. Here we wanted to highlight the possible assessment of nanotoxicological investigation in a 3D printed cell line OBSTs that can last for more than 3 weeks in culture without cell passage. Organoids-based scaffolds support the alternative to animal studies for longitudinal study following the 3R’s especially for the “Reduction” of animal testing because we can follow cell lines for several days, and mimic chronic conditions in-vitro. We modified the conclusions paragraph (see A7 below)

Q5. The most significant mechanism of action of Organoid-Based Scaffolds should be described and noticed more emphatically. Authors should discuss whether the use of Organoid-Based Scaffolds represents a solid alternative to existing test systems.

A5. We agree with the reviewer and we add a new sentence in the discussion section at Page 10, Line 289: “The indication that 3D OBSTs do not require cell passages and can survive in culture for >21 days may suggest adopting 3D for long-tem investigation in both field nanotoxicology and nanomedicine.”

Q6 Please add below studies to your manuscript in discussion section and also please discuss about possible role of this system in toxicity investigation of nanomaterials on cell and organells.

DOI: 10.7150/ntno.78611

DOI: 10.1016/j.jksus.2022.102340

DOI: 10.1155/2021/1520052

A6. Done as suggested

Q7. Conclusions should reaffirm the fundamental contribution of this paper.

A7. The conclusion paragraph has been rewritten.

“Our study demonstrates, the potential of 3D bioprinting as a promising technology for nanotoxicological investigation. The unique benefits of this novel additive manufacturing technology for cell culture include the long-term culturing of cell lines due to the reduction of oxidative stress over time, the viability of 3D printed cells once embedded in the hydrogel and the absence of time-passages from the operator. These benefits allow for a more accurate and comprehensive investigation of the effects of nanoparticles on living cells, as well as the potential for more efficient and cost-effective experimentation. Additionally, our research highlights the significant dissimilarities between 2D vs 3D data which suggest the need for a revisiting tactic in the fields of nanotoxicology and nanomedicine to account for potential effects on cell morphology and cell-cell interaction in a 3D environment. Ultimately, we believe that our technology can contribute to the development of safer and more effective nanomedicine, as well as provide a valuable tool for researchers in the field of nanotoxicology.”

Reviewer 4 Report

The topic is interesting, but some comments should be justified as follows:

1. Abstract: Some quantification data should be added to strengthen the abstract. The authors should state which materials they used to develop organoid-based scaffolds.

2. Many studies discussed the in vivo nanotoxicology using insects rather than rodent and rabbits and they obtained reliable and fruitful data utilizing cheap and available hosts without any limitations. In the light of this fact, please elucidate the importance of your study. Please use these articles to justify this point (https://doi.org/10.3390/antiox12030653; https://doi.org/10.1016/j.cbi.2022.110166).  

3. Results: Fig. 1 is not visible at all to assess, please replace it. Scale bars should be added in the figure incorporated in the SI.

4. Concerning oxidative stress: it is not sufficient to evaluate the lipid peroxidation. The authors should evaluate reduced glutathione (GSH) and metallothionein as crucial indicator for oxidative stress.

5. Discussion: it should be improved and updated with recently published articles.

6. Methods: which test did you for the normal distribution of the data?

7. Conclusion: please propose the future plan or general future perspectives based on the findings obtained in this study.    

Author Response

The topic is interesting, but some comments should be justified as follows:

We thank the reviewer for her/his valuable suggestions aimed at improving our manuscript

Q1. Abstract: Some quantification data should be added to strengthen the abstract. The authors should state which materials they used to develop organoid-based scaffolds.

A2. We thank the reviewer, and we followed his/her suggestions by adding some quantification in the abstract. “The cell number significantly increased on the 14th day in the 3D environment, from 5×105 to 1.27x106), showing 91% of lipid peroxidation reduction over time and minimal cell death observed throughout 21 days.”

The materials used for OBST are stated in the Results section, and the Method section (4.4.3)

Q2. Many studies discussed the in vivo nanotoxicology using insects rather than rodents and rabbits and they obtained reliable and fruitful data utilizing cheap and available hosts without any limitations. In light of this fact, please elucidate the importance of your study. Please use these articles to justify this point (https://doi.org/10.3390/antiox12030653; https://doi.org/10.1016/j.cbi.2022.110166).  

A2. We agree with the reviewer and we add those references in the introduction ((the last included silver nanoparticles in insects).

Q3. Results: Fig. 1 is not visible at all to assess, please replace it. Scale bars should be added in the figure incorporated in the SI.

A3, We apologize about that. Figure 1 was in a good resolution prior to the conversion to PDF. We replace Figure 1 with the High-Res version.

Q4. Concerning oxidative stress: it is not sufficient to evaluate the lipid peroxidation. The authors should evaluate reduced glutathione (GSH) and metallothioneina  as crucial indicator for oxidative stress.

A4. We understand the reviewer point; however we are confident to adopt TBARS as an index of lipid peroxidation induced by ROS in mammalian cells (See Savi et al . doi: 10.1186/s12989-014-0063-3, Rossi et al doi: 10.1186/s12989-014-0063-3, Rossi et al doi: 10.1016/j.envpol.2021.117163). ROS has a short half-life and because of the nature of our experiment, we have preferred TBARS evaluation (damage induced by ROS) with respect to other evaluation.

We have evaluated ROS (data not shown in the manuscript, see FACS figures below) of 2D (pink) vs 3D (red) after 24h; we did not detect any significant differences suggesting the OBST dos not increase ROS production (also confirmed by TBARS analysis).

Q5. Discussion: it should be improved and updated with recently published articles.

A5. Done as Suggested

Q6. Methods: which test did you for the normal distribution of the data? 

 A6. Normal distribution was checked by the Kolmogorov–Smirnov test. We add this information in the statistical analysis paragraph.

Q7. Conclusion: please propose the future plan or general future perspectives based on the findings obtained in this study.    

A7. We agree with the reviewer and we reformulate the Conclusion paragraph:

Our study demonstrates, the potential of 3D bioprinting as a promising technology for nanotoxicological investigation. The unique benefits of this novel additive manufacturing technology for cell culture include the long-term culturing of cell lines due to the reduction of oxidative stress over time, the viability of 3D printed cells once embedded in the hydro-gel and the absence of time-passages from the operator. These benefits allow for a more accurate and comprehensive investigation of the effects of nanoparticles on living cells, as well as the potential for more efficient and cost-effective experimentation. Additionally, our research highlights the significant dissimilarities between 2D vs 3D data which suggest the need for a revisiting tactic in the fields of nanotoxicology and nanomedicine to account for potential effects on cell morphology and cell-cell interaction in a 3D environment. Ulti-mately, we believe that our technology can contribute to the development of safer and more effective nanomedicine, as well as provide a valuable tool for researchers in the field of nanotoxicology.

Round 2

Reviewer 2 Report

The manuscript has been corrected according to my suggestions. 

Reviewer 4 Report

The authors addressed all the previous claims carefully; thus, I recommend accepting the current version of the manuscript.